# Direct Fluorination as Method of Improvement of Operational Properties of Polymeric Materials

**DOI:** 10.3390/polym12122836

**Published:** 2020-11-28

**Authors:** Nikolay A. Belov, Alexander Y. Alentiev, Yulia G. Bogdanova, Artem Y. Vdovichenko, Dmitrii S. Pashkevich

**Affiliations:** 1Engineering Center, Tomsk Polytechnic University, 30, Lenin Avenue, Tomsk 634050, Russia; alentiev@ips.ac.ru (A.Y.A.); pashkevich-ds@yandex.ru (D.S.P.); 2A.V. Topchiev Institute of Petrochemical Synthesis, Russian Academy of Sciences, 29, Leninskii Prospect, Moscow 119991, Russia; 3Chemical Department, M.V. Lomonosov Moscow State University, GSP-1, Leninskie Gory, Moscow 119991, Russia; yulibogd@yandex.ru; 4N.S. Enikolopov Institute of Synthetic Polymeric Materials, Russian Academy of Sciences, 70, Profsoyuznaya, Moscow 117393, Russia; vdartem@ya.ru; 5Institute of Applied Mathematics and Mechanics, Peter the Great St. Petersburg Polytechnic University, 29, Polytechnicheskaya, St. Petersburg 195251, Russia

**Keywords:** fluorination, polymer, surface properties, mechanical properties, wetting, Young modulus, elongation, permittivity, surface conductivity

## Abstract

Direct fluorination of polymers is a widely utilized technique for chemical modification. Such introduction of fluorine into the chemical structure of polymeric materials leads to laminates with highly fluorinated surface layer. The physicochemical properties of this layer are similar to those of perfluorinated polymers that differ by a unique combination of chemical resistance, weak adhesion, low cohesion, and permittivity, often barrier properties, etc. Surface modification by elemental fluorine allows one to avoid laborious synthesis of perfluoropolymers and impart such properties to industrial polymeric materials. The current review is devoted to a detailed consideration of wetting by water, energy characteristics of surfaces, adhesion, mechanical and electrical properties of the polymers, and composites after the direct fluorination.

## 1. Introduction

Fluorine-containing compounds have found a wide application in many fields of human activity. Despite a lack of natural organofluorines [1], industrial fluoroorganic derivatives are applied as specific medicines [2,3,4], agrochemicals [5], refrigerants [6], in the aerospace industry as materials with low densification temperature and high thermal stability (>300 °C) (oligomeric and polymeric perfluoroalkylene ethers) [7,8], etc. Huge attention is also paid to fluorine-containing and perfluorinated polymers [9]. They retain a separate niche among others conventional polymers because of a unique combination of properties such as excellent chemical resistance, permittivity, flame retardancy (owing to strong energy of C-F bond (485 kJmol^−1^) and weak polarizability of fluorine atom), hydro- and oleophobicity, weak adhesion and low cohesion due to low energy of intermolecular interactions with fluorine-containing groups [9,10]. The set of the properties allows applying the perfluorinated polymers as materials for wire and cable insulators, pipes and tubing, seals, resistant coatings, optical fibers, membranes, etc. [9,10,11,12,13,14,15].

However, the production of perfluorinated polymers faces significant difficulties related to the multi-stage and complicated synthesis of monomers, high prices of chemicals and solvents, and frequently low reactivity of the fluorinated monomers in polymerization [16]. Fortunately, most of the mentioned properties of the perfluorinated polymers can be achieved when a thin, highly fluorinated layer covers polymeric materials. Such highly fluorinated covering can be formed via physical coating of the material [17] or its chemical treatment (polymer-analogous reactions). The latter way of modification includes (i) chemical post-treatment through available functional groups [18,19,20,21], (ii) plasma-chemical treatment in the presence of fluorine-containing volatile compounds (tetrafluoromethane, hexafluoropropylene, hexafluorobenzene, etc.) [22], and (iii) direct fluorination [23].

The direct fluorination of polymeric articles (films, particles, fibers) has been repeatedly demonstrated to be an effective method for fine tuning of surface properties and related functional characteristics of polymer materials. Fluorination techniques allow one to obtain modified fluorine-containing surface layers with an adjustable thickness from 0.01 to 10 microns [23,24,25,26,27], chemical composition [23,28,29,30], and surface texture [28,31,32,33,34] while material bulk properties remain unchanged. An improved wetting of reinforcing elements with a polymer matrix increases the adhesion between the components of composite materials [30,35,36,37]. It reduces the concentration of defects at the interface [38,39], which leads to improved mechanical properties. Optimization of the wetting ability of polymer surfaces by fluorination is promising for use in offset printing [30,40,41], ensuring the protective properties of polymer materials [29,42,43,44,45,46], membrane [27,47]; and sorbents [38] performance. Direct fluorination of polymers also results in the formation of deep and shallow charge traps on a surface that improves dielectric [48,49,50,51], piezoelectric [52,53] properties, surface conductivity [54,55], breakdown [48,56], and DC flashover [57,58,59,60,61,62] voltages.

The necessity of the current review depends on several circumstances. The previous observations by Lagow et al. [63], Kharitonov et al. [23,40,64], Tressaud et al. [65,66] had been published more than a decade ago and had been concentrated on particular aspects of direct fluorination. So, Lagow et al. summarized own previous investigations on direct fluorination of various polymers (polyolefins, polyethers, nitrogen-containing ladder polymers) with the common characterization of the final products [63]. Kharitonov and Tressaud et al. thoroughly reviewed the fluorination kinetics, characterization of chemical structure of the fluorinated layers, and some aspects of application of the final products [23,40,64,65,66].

The current review of the direct fluorination of polymers does not cross the previous ones. Oppositely, it supplements them by focusing on a detailed consideration of properties of the final fluorinated polymeric materials that were poorly discussed previously, namely adhesion and surface energy (Section 2), mechanical properties of the fluorinated polymers and composites (Section 3), and electrical properties (Section 4).

## 2. Surface Properties of Polymers after Direct Fluorination

### 2.1. Wetting Method: The General Information

An informative and express method for monitoring the surface condition as a result of chemical modification is the wetting method, i.e., measuring contact angles of liquids in contact with a solid under various conditions [41,67,68,69]. This technique has been found in more than half of publications devoted to the direct fluorination of polymers.

The contact angle value is determined by the Young equation [68]:cos *θ* = (*γ*_SV_ − *γ*_LV_)/*γ*_SL_,(1)
where *γ*_SV_, *γ*_LV_, and *γ*_SL_ are the specific free surface energies of the interfaces of the phases involved in wetting.

The value of the contact angle is sensitive to chemical composition of the surface (the presence of terminal functional groups on it) [68] and its microrelief [69]. According to Ferguson & Whitesides [70], the wetting method provides information about the state of the external surface layers of solids at distances of 5–10 Å from the geometric interface to the depth of the phase, which corresponds to the local structure of the interfacial surfaces according to De Gennes (≈3 Å [69]). Since the range of the long-distance van der Waals forces is 30–300 Å [69], one can expect that the analytical depth of the method is tens of Å.

The experimental determination of the contact angle is usually performed by the sessile drop method (Figure 1). As a rule, the advancing contact angles are measured: i.e., the contact angles of test liquid drops on the surface investigated (Figure 1a); the angle is determined either goniometrically using a horizontal microscope, or using an approximate solution of the Laplace equation [68], which describes the shape of a drop in a gravitational field; the latter method is the basis of modern devices for determining the contact angle. The Laplace equation can also be used if the polymer object is a fiber in contact with the test liquid, as it was done before in the work of Cheng et al. [71]. Determination of the edge angle through capillary rise [72] and the Wilhelmy plate [36] methods are used in single studies of changes in the surface properties of polymers as a result of fluorination.

The simplest parameter characterizing a surface in terms of its hydrophilicity is the value of a contact angle of a water drop on it, more precisely, advancing contact angle *θ_a_*_(H2O)_. This parameter allows one to trace such an important property of the surface as its hydrophilicity (*θ_a_*_(H2O)_ < 90°) or hydrophobicity (*θ_a_*_(H2O)_ > 90°), i.e., to estimate in the first approximation the change in the functional properties of the polymer surface as a result of fluorination [32,73,74].

Separately, the phenomenon of superhydrophobicity, which can be observed as a result of fluorination of the polymer, should be noted. This is the property of the material to completely repel water, provided by the complex effect of hydrophobization of the surface and changing its microrelief. Nevertheless, when considering superhydrophobicity, the case *θ*_1_ = 180° cannot be realized in an air medium since the following is deduced by the Young equation for the equilibrium contact angle
cos *θ* = 2*W_a_*/*W_k_* − 1,(2)
where *W_k_* = 2*∙γ*_SL_ is the work of cohesion of liquid, *W_a_* = *γ*_SV_ + *γ*_LV_ − *γ*_SL_ is the work of adhesion of liquid to a solid.

The value of the water contact angle measured under receding conditions (when an air bubble is brought to the surface of a sample submerged in water, Figure 1b) may differ from *θ_a_*_(H2O)_. In this case, there is a static or ordinal hysteresis of contact angles, the value indicates the degree of roughness and/or chemical non-uniformity of the surface [68,69].

The wetting method is very useful in predicting changes in the adhesive properties of polymers as a result of surface fluorination. Thermodynamic characteristic of the adhesion between an adhesive (a) and a support (s) is the work of adhesion *W*_a_ = *γ*_a_ + *γ*_s_ − *γ*_as_, determined by the values of the total specific free surface energy of the contacting phases (*γ*_a_ and *γ*_s_) and their interfacial energy (*γ*_as_) [68]. Thus, the value of the specific free surface energy of polymer at the interface with air *γ*_a_ (*γ*_SV_ in Equation (1)) is more informative parameter for predicting the functional characteristics of polymer materials than the water advancing and receding contact angles. It should be noted that this value determines not only the adhesion of polymer materials, but also the influence on its adsorption characteristics and mechanical strength in contact with liquid media [40,75,76,77].

The possibility of determining the *γ*_SV_ of polymer by wetting method follows from the Young equation (Equation (1)).The specific free surface energy can be calculated from a macroscopic model of the thermodynamic state equation using experimental values of water contact angles, which was proposed by Neumann et al. [78] and developed by Chibowski et al. [79]. For these calculations, it is sufficient to know the contact angle of water on the polymer surface. Another way to estimate the value of *γ*_SV_ is the parachor [80,81,82]—the value that relates *γ*_SV_ to the molar volume of the polymer, which, according to the concept of Sugden [82], can be calculated by the group contribution method. However, such methods for analyzing patterns of changes in the specific free surface energy of polymers during fluorination are rarely used [83,84,85].

The molecular wetting theory of Fowkes [85] applied to low-energy polymer surfaces allow ones to calculate the dispersion *γ*^d^_SV_ and polar *γ*^p^_SV_ components of the specific free surface energy, the values of which serve as a response of the intensity of intermolecular interactions in the volume of the polymer phase. Aspects of the development and current state of the molecular theory of wetting are fully reflected in the reviews of Kloubek [86] and Sharma & Hanumantha Rao [87].

To determine energy characteristics of fluorinated polymer objects, as a rule, a two-fluid method is used, calculating *γ*^d^_SV_ and *γ*^p^_SV_ using the advancing contact angles of a pair of test liquids, one of which is water, the second is methylene iodide or formamide [88,89]. Calculations are usually performed using the Owens-Wendt approach [90], less often the Wu approach is used [91]. Sometimes, but more rarely, a set of test liquids is used [30,91].

Knowledge of *γ*^d^_SV_ and *γ*^p^_SV_ change allows one to directly regulate the adhesion of components in processes of composite materials preparation [92], and coatings application and susceptibility to dyes [23,41]. Despite this, many researchers limit analysis of these *θ_a_*_(H2O)_ experimental data to the qualitative evaluation of the degree of fluorination.

### 2.2. Changes in the Hydrophilicity/Hydrophobicity of the Polymer Surface as a Result of Fluorination

Replacing of H atoms with F atoms during fluorination can be expected to result in the creation of a Teflon-like surface, for which *θ_a_*_(H2O)_ = 118–120° [93]. However, different patterns of change in water wetting after fluorination are observed for polymers containing polar functional groups and hydrocarbon polymers.

Mild fluorination of hydrophilic polymers with a mixture containing an inert gas (He or N_2_) gives rise to the cleavage of hydrogen bonds in the surface layers, the transition of the terminal polar groups (–OH, C=O, –COOH, –NH_2_) to the surface and its subsequent hydrophilization (Figure 2) [71,84]. This technique is used to improve the adhesion of hydrophilic components of composite materials at the prepreg stage in order to improve mechanical properties (see Section 3).

At long times of fluorination and a higher concentration of fluorine in the mixture, the content of fluorine in the surface layer increases, which leads to hydrophobization of the surface. Thus, an increase in the water advancing contact angles were detected for the fluorination of plant fibers [74,94,95] and composite materials filled with wood flour [96,97]. For plant fibers, optimal fluorination conditions have been established to ensure the maximum degree of hydrophobicity of the material [74] since the oxidative destruction of macromolecules occurs during prolonged exposure to the aggressive agent. It is accompanied by an increase in the hydrophilicity of the material. In the case of a wood flour-based composite material [97] that initially has limited wetting with water, no wetting inversion occurs for the samples after fluorination (the transition from hydrophilicity to hydrophobicity). In this regard, the water absorption of such composites decreases insignificantly [96].

Changes of wetting ability for the hydrophobic polymers and the polymers limitedly wetting with water depend on the conditions of fluorination. For example, fluorination in an inert atmosphere leads to hydrophobization of polyethylene [98] due to the occurrence of fluorine-containing groups (Figure 3). While the presence of oxygen in the reaction mixture in parallel with the substitution of hydrogen to fluorine leads to the formation of polar groups on the surface –CHF(C=O)SNF– and –SNF(C=O)CH_2_– or –CHF(C=O)O–, –C(O)F, –C(O)OH or –CHFC(O)O– [41] having a high affinity to water. The set of the detected functional groups for the fluorination of polyethylene in the presence of oxygen is presented in Figure 3. Thus, oxyfluorination leads to hydrophilization of the polymer surface [30,36,98,99,100,101]. The hydrophilization effect was also observed for the oxyfluorination of polypropylene-based materials [30,73,102].

In the case of materials with a considerable microrelief of the surface, information is needed about whether the wetting mode is homogeneous (a liquid is continuously in contact with a solid surface) or heterogeneous (liquid contacts only the vertices of the microrelief of a solid surface) [41,104]. It is possible to estimate the wetting mode by comparing the size of surface irregularities with the capillary constant of the liquid [41]. The effect of roughness on contact angles in a homogeneous wetting mode (the Wenzel–Deryagin equation) is discussed in [41,43]; and in a heterogeneous mode (the Cassi–Baxter equation) is discussed in [32].

The result of fluorination of crosslinked polydimethylsiloxanes depends on their chemical structure: the initially hydrophobic surface of PDMS may become (i) superhydrophobic [54] and hydrophilic (with improved wetting ability with alcohol solutions) [103].

Information about the wetting mode is relevant for superhydrophobization of polymer surfaces due to the complex effect of increasing the content of fluorine in the surface layer and surface morphology features that can be created by directional etching [41] or by introducing fillers into a polymer [34]. Such surfaces were obtained by oxygen-free fluorination of silicone rubbers (*θ_a_*_(H2O)_ = 143.7°) [42,105,106] and carbon fiber, which is a component of a composite material based on polyvinylidene fluoride (*θ_a_*_(H2O)_ = 153°) [34]. Particular attention should be paid to the effect of “sticky superhydrophobicity” of polybutylene terephthalate fiber mats as a result of liquid-phase fluorination [32]: when a drop of water is not separated from the surface by gravity even in the case of non-wetting surface (*θ_a_*_(H2O)_ = 156°). This effect is realized due to the Cassi state: the three-phase (liquid/solid/air) contact line is continuous at the micro-scale, but it turns out to be discontinuous at the nanoscale. Note that the water contact angle for the initial samples is quite large (*θ_a_*_(H2O)_ = 126°), which is evidently due to the originally developed texture of their surface. Fluorination of aramid fabric provides its omniphobic properties [39].

It should be noted that more information can be extracted from water wetting experiments than simple confirmation of the surface hydrophilicity change. So, the analysis of the change in the value of the ordinal hysteresis of water contact angles (the difference in the values of advancing and receding contact angles—see Figure 1) may shed light on the energy uniformity of the initial surface and its change as a result of modification of the polymer surface layer. This information can be useful at the sampling stage in order to obtain reproducible results and correct interpretation of the data. However, hysteresis phenomena during wetting of modified surfaces are discussed only in a few papers [72,98,105,107]. Le Roux et al. [72], F. J. du Toit & Sanderson [98] associate an increase in water wetting hysteresis with an increase in the roughness of the surfaces of polyphenylene oxide and polypropylene caused by their etching as a result of fluorination, Gekas et al. [107] indicate a correlation between the hysteresis and porosity for commercial ultrafiltration membranes based on cellulose triacetate and polysulfone. On the contrary, Gao et al. [108] found a smoothing of the surface microrelief of fluorinated elastomers and a decrease in the hysteresis of water contact angles as a result of additional fluorination, which made it possible to unambiguously associate an increase in *θ_a_*_(H2O)_ with an increase in the fluorine content in the polymer surface layer.

Useful and informative for estimating the degree of surface modification *φ* is the Cassi–Baxter equation of the theory of wetting of heterogeneous surfaces [106]:cos*θ* = *φ*∙cos*θ*_1_ + (1 − *φ* )∙cos*θ*_2_,(3)
where *θ* is the advancing contact angle of water determined experimentally, *θ*_1_ and *θ*_2_ are the contact angles on the fully modified and original surface, respectively, taken from the literature data.

As an example of the interpretation of experimental data within this equation, Table 1 shows the estimated values of the degree of modification (hydrophobization or hydrophilization) of various polymer objects via fluorination and oxyfluorination using literature data. The values *θ*_1_ were chosen to be 120° for a fully fluorinated hydrophobic surface [93] and 0° for a surface that is completely wetted with water [68]. In accordance with Equation (2), when the superhydrophobicity effect is displayed *θ*_1_ does not reach the value of 180°, so for calculations in case of superhydrophobic surfaces, it was assumed that *θ*_1_ = 170° is the maximum value of the water contact angle found in the literature [109].

Interpretation of the results on water wetting of the fluorinated polymers surfaces in the context of Equation (3) is not found in any of the works, although the calculation of the *φ* value allows one to rapidly track the completeness of the fluorination reaction depending on the conditions. An attempt was made to analyze the degree of change in the hydrophilicity of natural fibers as a result of fluorination in terms of a percentage increase of the water contact angles (the maximum increase was 92%) [94]. However, it should not be forgotten for such estimation that the thermodynamic value describing wetting is the cosine of contact angle in accordance to Young equation (Equation (1)).

### 2.3. Adhesion and Specific Free Surface Energy of Fluorinated Polymers

Fluorination allows regulating such physical and chemical factors of adhesion assurance as density [111] and energy [112] of intermolecular interactions at the interface, mechanical adhesion of an adhesive and a substrate [113], stress relaxation in the transition interface layer [111], e.g., by reducing defects at the interface [114]. Let us recall that the *γ*_SV_ value is an important parameter for ensuring polymer adhesion to the support, so it is important to understand the effect of fluorination on this value and its polar and dispersion components.

In general, hydrogen substitution with more electronegative atoms (fluorine or oxygen) leads to a decrease in dispersion interactions and, subsequently, to a decrease in the dispersion component of the surface energy. The contribution of the polar component depends on the value of dipole moments localized in the macromolecule, which increases with the asymmetric addition of electronegative atoms to the polymer chain, as rightly pointed out by Le Roux et al. [72].

Since the energy of polar intermolecular interactions exceeds the energy of dispersion interactions [68], the response of the intensity of intermolecular interactions at the interface is the value of the polar component of the specific free surface energy *γ*^p^_SV_, which is small for most polymers with practical applications (Table 2) [72].

An effective way to increase *γ*^p^_SV_ of polyolefins is oxyfluorination. Depending on the reaction conditions, *γ*^p^_SV_ increases from 0 to 22–42 mJ/m^2^ for PE [30,36,37], from 2 to 36 mJ/m^2^ for PP [98]. Meanwhile, the specific free surface energy of polyethylene and polypropylene increases to 72 and 56.5 mJ/m^2^, respectively [41]. This provides a stronger adhesive interaction of polyolefin fibers with oligomeric binders and their susceptibility to dyes [23]. Exposure to fluorine without addition of other gases provides a complex effect of changing the chemical composition of the surface and increasing its roughness to increase the surface energy of polyolefins [91]. It should be noted that the value of *γ*^p^_SV_ for non-polar polymers is more variable at oxyfluorination than in the case of polar polymers [115].

To increase *γ*^p^_SV_ of polymers containing polar groups, fluorination in an inert atmosphere is effective since it forms terminal polar groups on the surface as a result of the destruction of intramolecular hydrogen bonds of these polymers. An example is the hydrophilization of aramid fibers, the fluorination of which is studied more widely. Aramid fibers are initially hydrophobic (*θ_a_*_(H2O)_ = 113–117°), and as a result of mild fluorination, even wetting inversion is not achieved (*θ_a_*_(H2O)_ = 98–103°), but this exposure leads to an increase in the surface energy of the fibers from 13 to 25 mJ/m^2^ and provides to increase of its adhesion to epoxy resins which leads to increase of strength of composite material [71,84]. It is very clearly illustrated in Figure 4 given by Cheng et al. [84].

In addition to breaking intramolecular hydrogen bonds, fluorination activates the surface due to the appearance of reactive functional groups and increases the probability of chemical free-radical reactions. These factors are used both in the curing reaction of oligomeric binders to enhance the adhesion interaction [84] and for covalent grafting of coupling agents in parallel with the polymerization of the matrix [111] in order to counteract the occurrence of stress in the interface or to ensure dispersion of the filler in the matrix [34]. Fluorination-activated graft-polymerization is also possible for polymers that have low reactive functional groups in the polymer chain. Thus, grafting styrene and acrylonitrile to the surface of polyethylene provides an anti-adhesive effect, and grafting aniline and styrene onto PP improves its susceptibility to dyes [45]. Prospects for the strategy of functionalization of the surface of ultra-high molecular weight polyethylene (UHMWPE) by activating its surface with fluorine are noted by Li et al. [116].

The contribution of the dispersion component of the specific free surface energy *γ*^d^_SV_ to the adhesion interaction cannot be underestimated. The *γ*^d^_SV_ value related to the density of the polymer in the surface layer [82] can serve as an indicator of the density of adhesive bonds at the interface between an adhesive and a substrate. Fluorination increases the density of polymers [83,117], but the trends in the change of *γ*^d^_SV_ resulting from fluorination differ significantly for the groups of polar and non-polar polymers [37,54,101,106,118,119]. This may be due to the loosening of the polymer surface layer as a result of increasing roughness under different fluorination conditions. An example of the synergistic effect of increasing the fluorine content in the surface layer and the development of the microrelief of the surface of aramid fabric on the dispersive mode of specific free surface energy is its ultra-low *γ*_SV_ value (6 mJ/m^2^), obtained by Jeong et al. [39]. In contrast, Zhu et al. [120] showed that 20% increase in *γ*^d^_SV_ was indicated for mild fluorination of nitrile-butadiene rubber, while its *γ*^p^_SV_ was insensitive to the exposure.

An increase in the specific surface area of fibers as a result of fluorination is an additional factor in increasing of the adhesion of composite materials components [35,121]. Mild fluorination improves adhesion by healing defects at the interface between the components of composite materials [38,39,97], enhancing intermolecular interactions in the polymer surface layer [114,122], eliminating weak boundary layers [114,123], and reducing the polarity difference of the contacting phases [124]. Densification of the surface layer of the polymer is also possible due to crosslinking [114] and an increase of the crystallinity [122].

It should be noted that information about the stability of free surface energy characteristics over time is essential for predicting the duration of the material’s functioning. Despite this, a study of behavior of the energy characteristics at the aging of fluorinated surfaces is found in a small number of works [30,41,66,112]. Considering aging, fluorination provides more stable surface properties of polymers compared to plasma-chemical surface treatment, as shown by Nazarov et al. [41].

Summarizing, the direct fluorination of polymeric materials is an effective method to tune their surface properties. Depending on fluorination conditions, the surface can become hydrophobic (it is similar to the generation of protective coating) and hydrophilic (by the generation of polar groups due to the chemical modification). Another promising option of the technique is a combination of the hydrophobization with the simultaneous adjustment of the microrelief in order to form superhydrophobic surfaces. Fluorination also allows fine adjustment of polymer adhesion, which is a key factor in controlling the mechanical strength of polymer composite materials. It is also pointed out that the fluorination does not affect the bulk properties of the polymeric materials touching upon their surface layers that are responsible for the adhesion characteristics.

## 3. Mechanical Properties of Polymers and Composites after Direct Fluorination

For the practical application of polymer films, coatings, fibers, filled polymers and composites, the most important target technical parameters are the mechanical characteristics of polymer materials. Direct fluorination of polymers leads to the formation of a bilayer structure, improved adhesion characteristics, and surface layer stiffness, which, in general, increases the elastic modulus, and the tensile strength of the polymer material and declines the elongation at break.

So, gas-phase fluorination (15%v/v of fluorine in helium) of PETP films gives rise to the formation of a surface structure morphologically different from the virgin film [125]. When such films are stretched, the surface structure becomes similar to that observed after stretching of polymer films with a thin metal coating [126]. Namely, the surface layer of the polymer breaks up into fragments of a similar size (Figure 5). At the same time, deformations above the glass transition temperature of PETP form a regular folded relief. The authors attribute this phenomenon to higher values of mechanical parameters of the fluorinated layer and estimate its strength at break (*σ*) as 60.7 MPa when it is deformed at room temperature and as 3.8 MPa at 90 °C. Simultaneously, the strength value of the fluorinated surface layer is close to the similar strength value of the layer that occurs on the surface of a PETP film after cold plasma treatment [127], which indicates the presence of a large number of crosslinks in the surface layer. An increase in the stiffness of the surface layer also leads to a decrease in elongation at break (*ε*) up to 15–30% at room temperature and up to 8–10% at 90 °C.

The fragmentation of films under stretching is also observed for low and high-density polyethylene under similar conditions [128]. Namely, an increase in Young modulus (*E*) up to 50%–120% is associated with a decrease in *σ* up to 20–35%, and *ε* by 9–11% (Table 3). In this case, the calculation of the strength *σ* of the fluorinated layer shows an increase by 4–6 times for LDPE and by 20-30% for HDPE [129]. The thickness of fluorinated layer depends evidently on the time of fluorination and properties of the polymeric material and, correspondingly, on the fraction of the fluorinated layer.

For nitrile-butadiene rubbers [120], a sharp increase in the surface modulus (by 500%) is observed while the tensile and bending strengths change little (Table 3). At the same time, for O-rings and flat samples (Table 3), a negligible variation of *E* in the range of 6–8% is accompanied by a decrease in *σ* by 24–28% and in *ε* by 35–53% [44]. For cross-linked natural rubber [84], fluorination and oxyfluorination practically do not change the mechanical characteristics of the samples (*σ* = 26 ± 2 MPa, *ε* = 900–1000%). However, in the case of aging at 70 °C for a week, oxyfluorinated and fluorinated samples show a decrease in tensile strength by 50% and 70%, respectively. Such behavior of the rubber samples is also explained by the presence of a more rigid surface fluorinated layer, the contribution of which to the overall mechanical characteristics depends on the thickness of the sample and, correspondingly, on the time of fluorination.

Direct fluorination turns out to be a very promising way to modify the properties of polymer composites, giving them the characteristic high thermal stability, chemical resistance, and hydrophobicity, as well as low water absorption for fluorinated polymers [9]. The advantage of the method is the technological controllability of the film surface formation process due to the variation of the composition and pressure of the fluorination mixture as well as fluorination duration. One can adjust the thickness of the layer, its continuity, hydrophilic or hydrophobic properties of the surface (see Section 2), and can also change the adhesive interactions of the matrix and the filler, which leads to changes in their mechanical characteristics.

Surface modification of composites usually involves fluorination of the final composite material (Case I) and fluorination of the filler introduced into the polymer matrix (Case II). In the first case, mainly surface fluorination of the polymer matrix of the composite occurs, so the main contribution to the properties of the composite is made by changing the characteristics of the polymer matrix, and the regularities of changes in the properties of polymer films, fibers and composites during surface fluorination are common. In the second case, if the filler is a polymer fiber, the main influence on the properties of the composite is again made by the fluorinated surface layer of the polymer. When the filler is inorganic, for example, recently popular carbon nanotubes (CNTs), the fluorinated surface of the filler may significantly affect the properties of the final composite.

So, the formation of a hard surface fluorinated layer during fluorination of UHMWPE and its composites (Case I) containing graphite nanoplates, montmorillonite, molybdenum disulfide, and shungite as fillers [136,137] leads to improvement of the wear resistance of samples. An increase in the surface layer stiffness during surface treatment of UHMWPE fibers (Case II) by fluorination and oxyfluorination results in higher total Young’s modulus (Table 3) for its composites with LDPE matrix [130,131]. Such behavior was explained by stronger adhesion of the fluorinated fibers and matrix due to their better mechanical interlocking. A similar effect is utilized to fabricate composite materials based on the fluorinated UHMWPE filler and epoxy resin by Kudinov et al. [132], or thermoplastic polyurethane by Li et al. [103,138,139] (Table 3).

When structure of polymers is stabilized by a hydrogen bond network (such as cellulose [97,133], polyaramides [71], and polyoxadiazoles [140]), the surface fluorination results in disruption of the network. These changes lead to improved adhesion characteristics between the polymer filler (dispersed filler, or fibers) and binders based on epoxy [71] or polyester [97] resins. An increase of the mechanical characteristics of composites based on cellulose (wood flour) and polyester resins [97,133] is also observed (Table 3). The infraction of the hydrogen bond grid leads to the fact that the fiber contact strength with the epoxy resin (pull-out test) increases by 40% despite a slight decrease in the fiber strength [35].

For instance, non-modified Twaron fibers are easily deboned from the PP matrix during tensile fracture, creating holes is clearly seen in the micro-image (Figure 6a), while the fluorinated fibers are stripped off during the pull-out (Figure 6b). The former case corresponds to poor interfacial adhesion, and the latter does to better interfacial bonding. The effect of filler treatment results in the improvements of Young’s modulus and tensile strength (Table 3) for the number of reinforced composites based on modified polyaramide fibers (Twaron [24,88,141], Kevlar [134,142,143,144], or PBIA [145], as well as Kevlar fibers treated with hydrofluoric acid [135]) and polymer matrices (such as polypropylene (PP) [24,88,141], ethylene and propylene copolymer (EP) [134,142,144], polystyrene (PS) [143], vinyl ether [135] and epoxy resins [145]). In some cases, one observes the improvement of the thermal stability of the composites due to an increase in the glass transition temperature [135,143,145] or an increase in the melting point [141] and the degree of crystallinity [24,88,141].

The fluorination of inorganic fillers leads to a slight increase in the strength of the composite. Thus, the pre-fluorination of illite increases the tensile strength (Table 3) of polypropylene-based composites by only 7% compared to a composite based on non-fluorinated illite [124]. For composites based on epoxy resins, the introduction of fluorinated single-wall CNTs leads to an increase in the elastic modulus of the composite [146]. More complicated behavior is observed for fluorinated multi-wall CNTs by Blokhin et al. [113]. Namely, the mechanical characteristics of the final composites based on epoxy resins strongly depend on the fluorination conditions (Table 3) and CNTs concentration resulting in both increase and decrease of tensile and bending strengths of the composites [113]. At the same time, in order to increase the compatibility of fluorinated CNTs with epoxy matrices reinforced with Kevlar fibers, complex schemes for grafting polymers onto the surface of fluorinated fibers and modifying the surface of fluorinated CNTs with polyethylenimine are used by Lv et al. [145].

Thus, surface fluorination is one of the most important ways to regulate the mechanical characteristics of polymers and composites. The undoubted advantage of polymers and composites treated with elemental fluorine (over the improvement of the elastic modulus, and the tensile strength, etc.) is additional properties typical to fluoropolymers: increased chemical resistance (and in some cases heat resistance), reduced water absorption.

## 4. Electrical Properties of Polymers after Direct Surface Fluorination

Besides improving wettability, adhesion and chemical stability, direct fluorination is effective approach to modulate the electrical properties for use as an insulating material for transformers, high and extra-high voltage cables, gas-insulated switchgears, inverter-fed motors etc. For the successful use of polymer dielectrics, it is necessary to control the main cause of aging and breakdown—the distribution of space charge in polymer materials [147,148].

The primary cause of the electric breakdown is the space charge accumulation in polymeric dielectric by charge injection at the electrodes and the field-assisted thermal ionization of internal impurities [55,148,149]. But surface charge accumulation on the insulator causing flashover is a result of a dynamic unbalance between the two contributions: one is the current driven by the normal component of electric field, and another is given by the gradient of the current flowing along the spacer surface, driven by the tangential component of electric field. Therefore, it is desirable to minimize the charge accumulation by an increasing of the surface conductivity [150].

Charge traps in a polymer are formed by defects in amorphous and crystalline regions, chemical defects like macromolecular chain end or impurities, which have significant influences on charge carrier trapping, detrapping, transport, recombination, and space charge formation, etc. So, there are several competing factors for the alteration of polymer conductivity after fluorination.

On the one hand, since fluorine atoms are strongly electronegative and have a large atomic radius it is easy to capture electrons [147]. Also the substitution of H atoms by F atoms in the polymer chain results in shorter inter-chain distance, less amorphous space and more condensed molecular conformation, which therefore forms deeper electron traps, while holes, generally attracted by the intra-chain chemical defects with much stronger bonding have no clear tendency to change [57]. Therefore fluorination leads to the formation of deep electron traps in the surface layer, electric field at metal-insulator interface becomes lower, and the further charge injection from electrode is blocked by the screening effect of the trapped electrons charge layer [55,57,147].

On the other hand, the changes in the surface morphology after fluorination provide a large number of physical interfaces for shallow traps as well as an increasing the degree of scission [57] and fluorination also increases surface conductivity, which facilitates surface charge dissipation and suppress charge accumulation. It has been shown that direct fluorination can increase surface conductivity by several orders of magnitude (Table 4).

Such contradictory phenomena results from competition between the compositional change and the structural change, shown schematically in Figure 7. The structural change plays a dominant role over the compositional changes. Xie et al. reported [54] that fluorine treatment increases the surface conductivity, but at the same time annealing decreases the surface conductivity of the fluorinated PDMS sheets by reducing the number of the structural defects, i.e., shallow physical traps.

In addition, Zhou et al. [151] defines another factor reducing the flashover voltage – surface roughness, which increases by direct fluorination process. Field emission near the cathode creates seed electrons colliding with the dielectric surface and generates secondary electron emission. A plasma discharge then occurs within the local gas layer as the electrons ionize desorbed gas. High surface roughness blocks some emitted secondary electrons thereby moderating secondary electron emission avalanche.

These competitive processes - the appearance of fluorine atoms and structural transformations in polymers - can be traced in the evolution of electrical properties of polymers on the fluorination time. Direct fluorination of PE usually leads to significant increase in the surface dielectric constant (Table 4) due to the appearance of strongly polar groups, like C=O containing and some of the -CF_x_ groups, such as -C(O)F, -C(O)OH, -CHF-, -CF_3_, etc. [147,158,159]. The surface charge decay rate in PE does not improve, but rather slows down by fluorination [147,151,159,160], in contrast to polymers with the benzene ring or branches fracture in molecular chains [59]. For example, silicone rubber, first demonstrates a decrease in the dielectric constant, and then the increase to the initial values [51]. The same trend was found for permittivity in polyimide [48,49], conductivity and charge dissipation rate in PDMS [54], paper-based insulators [155,161,162], and epoxy-based composites [60,61]. The authors associate this phenomena with low electron polarizability of fluorine atom and an increase of the free volume due to the relatively large volume of fluorine compared with hydrogen (which reduces the number of polarizable groups per unit volume) at the first stage, and then with a growth of the number of broken Si-C, Si-O, C-H, C-OH bonds, aromatic imide groups and polymer chain scission under long time fluorination.

Comparing direct and plasma fluorination, one can conclude that plasma-fluorinated surface layer does not possess the same suppression effect as in direct-fluorination treatment. Large amount of homocharges is observed inside the bulk, which is not visible for the direct-fluorinated samples [154,163,164].

If the insulator is to be operated in a humid environment, absorbed water will affect the surface charge dynamic behavior and the breakdown voltage will be significantly reduced. Fluorination prevents water absorption due to the appearance of hydrophobic fluorine-containing groups and a decrease in surface free energy. Thereby, direct fluorination partially suppresses the effect of lowering the breakdown voltage [48,162].

Recent studies have shown the effectiveness of direct fluorination in modulating the dielectric properties like conductivity, permittivity, breakdown and flashover voltage of the surface layers of such large-scale produced insulating polymers as polypropylene [162], PE [55,147,149,158,159,160,165], PS [166], polyimide [48,49,167], epoxy resins [51,150,153,154,166], PDMS [42,54,168], oil-impregnated [161,162,163], and polypropylene laminated paper [162].

Moreover, to improve dielectric properties, direct fluorination is used in combination with the filling of polymers with inorganic nanoparticles. Simultaneous application of nano- filling and direct fluorination may lead to significantly higher DC breakdown and flashover strength [57,59,60,61,161,169].

Direct fluorination can increase piezoelectric properties of polymers. An and coworkers [52,53] showed that piezoelectric activity and thermal stability of the cellular polypropylene film can be significantly improved by the deeply trapped charge on the internal layers of the fluorinated sample. Therefore, direct fluorination can be meaningful for developing applications of the piezoelectric films as sensors in elevated temperature environments.

Application of fluorinated polymers in different electrochemical systems (e.g., polymer insulating and paint coatings, conducting polymers for corrosion protection, electrodes, polymer binders for electrodes in batteries, polymer electrolytes, sensors or electrocatalysts, membranes, battery cases, etc.) imposes new restrictions on the use of polymers that differ from traditional requirements for polymer stability [169]. Electrochemical reactions of a polymer with electrolyte or electrochemically transformed components of the system lead to its degradation and even to a premature failure. Pud et al. [169,170] reported that PVTMS and PPO after fluorine treatment is able to participate in electrochemical reactions and undergoes electrochemical reductive degradation. Fluorination leads to dramatically narrowing of the range of electrochemical stability potentials. Perfluorinated chains participate in direct electrochemical reactions with elimination of fluorine anions producing conjugated bonds in the macromolecules.

## 5. Concluding Remarks

The treatment of common polymer-based materials with elemental fluorine was demonstrated to be the effective approach for chemical modification of their surface. The generated highly fluorinated thin layers can behave as either hydrophobic or hydrophilic depending on the nature of polymer and fluorination conditions (the content of fluorination mixture and duration of treatment). These peculiarities allow one to increase adhesion of particles, fibers, fillers to polymer matrices and to produce composite materials with improved mechanical properties. The direct fluorination also modifies the electronic structure of the surface layer (the screening effect of deep electron traps inside the fluorinated layer near the electrode, the appearance of shallow traps near the surface), resulting in the enhancement of electrical breakdown, surface flashover performance, and conductivity of wide range polymers and polymer composites.

Moreover, the fluorination technique offers an opportunity for fabrication of superhydrophobic surfaces, the development of scientific background of which is topical task of material science. In addition, there is a key fundamental issue that remains, namely a verification of surface tensions γ_SL_ and γ_SV_, estimated in the framework of molecular theory, of wetting by contact angles based on modern approaches. Atomic force microscopy would have been a very useful tool in this case, but it is often utilized for the investigation of the microrelief of the fluorinated polymeric surfaces. These and other items might become the subjects of further investigations.

## Figures and Tables

**Figure 1 polymers-12-02836-f001:**
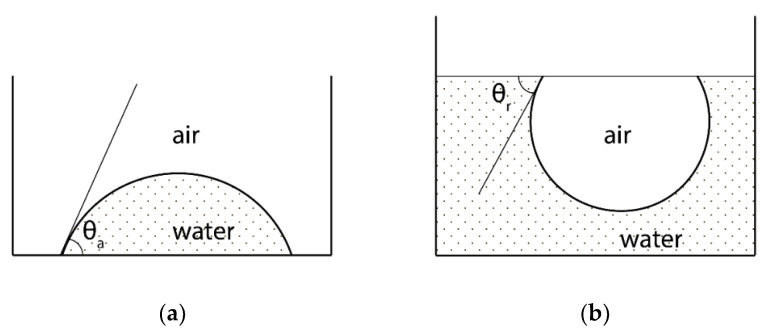
Scheme for measuring advancing *θ_a_* (**a**) and receding *θ_r_* (**b**) contact angles.

**Figure 2 polymers-12-02836-f002:**
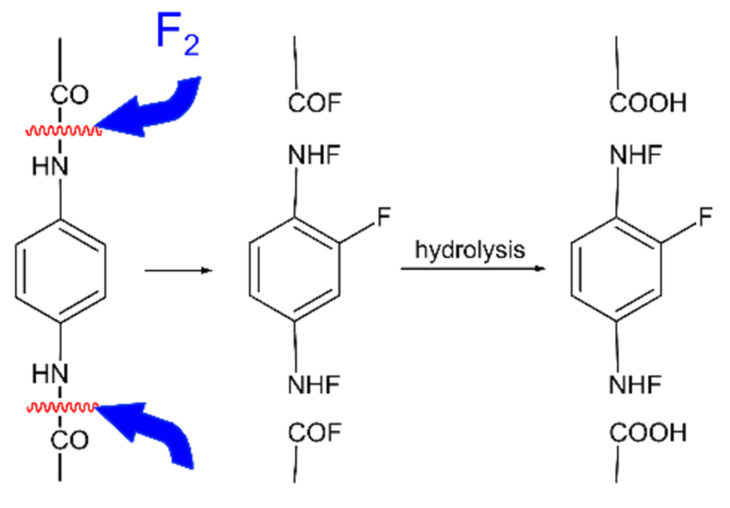
A scheme of interaction of hydrophilic poly(*p*-phenylene-benzimidazole-terephthalamide) with elemental fluorine and subsequent hydrolysis adopted from ref. [84].

**Figure 3 polymers-12-02836-f003:**
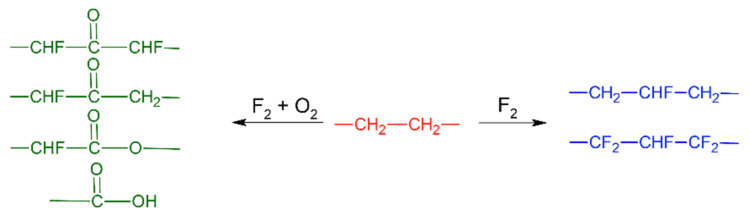
A scheme of direct fluorination and oxyfluorination of polyethylene [41,103].

**Figure 4 polymers-12-02836-f004:**
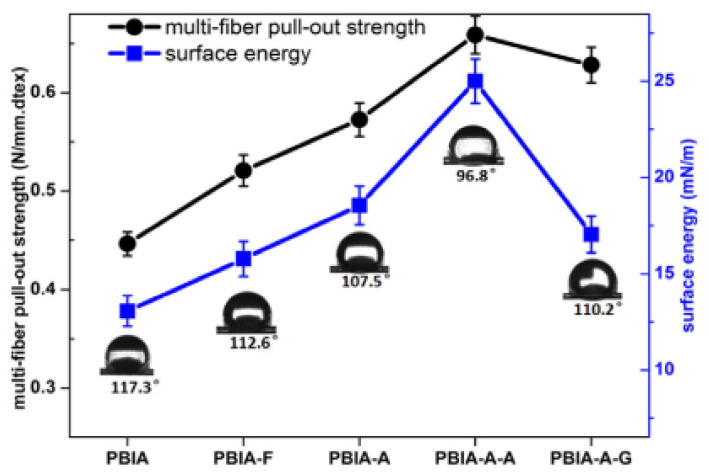
The relationship between different fiber’s surface energy and adhesive property of the PBIA-based composites: PBIA-F (fluorinated PBIA), PBIA-A (grafting Si-OH as end groups), PBIA-A-A (amine functionalization) and PBIA-A-G (epoxy functionalization). The figure is adopted from ref. [84].

**Figure 5 polymers-12-02836-f005:**
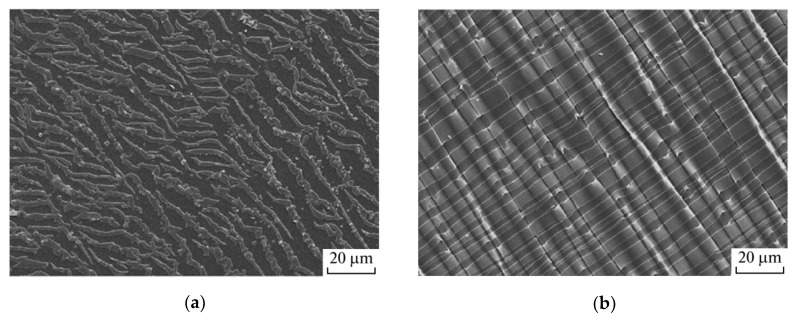
Scanning electron micrographs of PETP film specimens that were subjected to gas-phase fluorination for 30 min and then stretched (**a**) at room temperature until necking began and (**b**) at 90 °C by 100%. The SEM images were taken from ref. [125].

**Figure 6 polymers-12-02836-f006:**
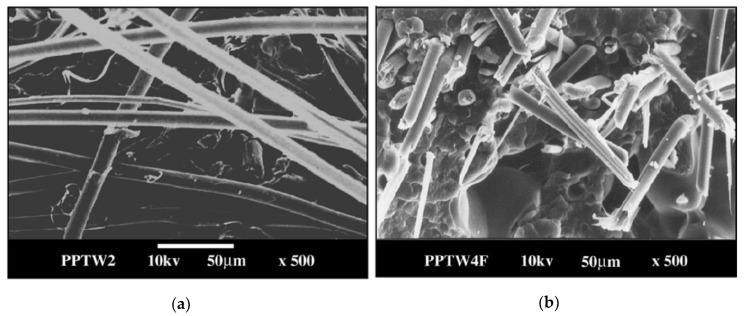
Scanning electron micrographs of broken PP based composites with non-modified (**a**) and treated with elemental fluorine (**b**) Twaron fibers. The SEM images are taken from reference [141].

**Figure 7 polymers-12-02836-f007:**
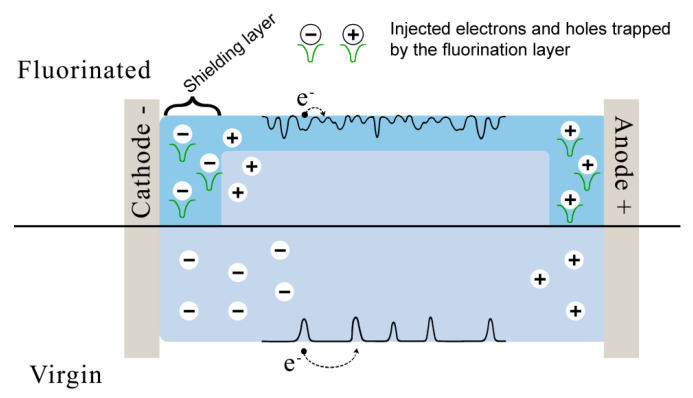
The schematic model of the effect of fluorination on space charges distribution and surface conductivity.

**Table 1 polymers-12-02836-t001:** Contact angles of water droplets and degree of chemical modification of polymer objects during fluorination according to Equation (3).

Fluorinated Object	*θ*_1_, Degrees	*θ*_2_, Degrees	*θ,* Degrees	*φ,* %	Ref.
Aramid fiber	0	113	103	12	[71]
	0	117	97	23	[84]
PP (+TiO_2_) (composite)	0	101	67	49	[30]
Wood flour + polyester (composite)	120	76	84	19	[97]
PP (non-woven fabric)	120	77	86	21	[99]
	0		64	28	
Wood fiber	120	51	98	68	[94]
	120	65	120	100	[74]
Cotton fiber	120	0	117	97	[95]
Aminated polyetherimide (film)	120	0	86	62	[110]
Polybutylene terephthalate (fiber)	170	126	156	82	[32]
Silicone rubber (plate)	170	110	144	73	[46]
Carbon fiber	170	110	153	85	[34]

**Table 2 polymers-12-02836-t002:** Specific free surface energy of polymers and its components, [*γ*] = mJ/m^2^ [72].

Fluorinated Object	Dispersion *γ*^d^_SV_	Polar *γ*^p^_SV_	Total *γ*_SV_
PE	32.0	1.1	33.1
PP	30.1	0	30.1
PVF	31.3	5.4	36.7
PVDF	23.2	7.1	30.3
PTFE	18.6	0.5	19.1
	18.4	1.7	20.1
PETP	37.8	3.5	41.3
	35.6	9.0	44.6
PMMA	35.9	4.3	40.2
	29.6	11.5	41.1

**Table 3 polymers-12-02836-t003:** Mechanical properties of fluorinated polymers, and composites.

Polymer Matrix	*Filler*	*Fluorination Procedure*	*E*^1^, *MPa*	*σ*^1^, *MPa*	*ε*^1^, *%*	*Ref.*
LDPE	-	15% F_2_ + He; 3 h	72.6/111.2	20.5/16.3	647/634	[129]
LDPE	UHMWPE—short fibers	5% F_2_ + He; 1 h (fibers)/20% F_2_ + He; 1 h (surface)	458/704/765	11.21/21.16/24.03	18.15/8.06/5.71	[130]
LDPE	UHMWPE—short fibers	10% F_2_ + He; 2 h	248/737	10.00/28.72	20.31/4.25	[131]
HDPE	-	15% F_2_ + He; 3 h	310/670	56/44	520/460	[128]
TPU	UHMWPE—particles of 50–70 microns	10% F_2_ + N_2_	-	12.5/22.3	523.5/892.1	[103]
TPU	UHMWPE—particles of 50–70 microns	10% F_2_ + N_2_/9% F_2_ + 14% O_2_ + N_2_	-	12.5/16.5/23.7	523.5/645.1/956.7	[103]
-	UHMWPE—fibers	5% F_2_ + He; 1 h	-	621/797	-	[132]
NBR	-	10% F_2_ + N_2_; 1 h	-	16/17	-	[120]
NBR	-	F_2_ + He; 5 h /100 °C	11.7/12.4	21.1/16.1	376.1/176.2	[44]
Norsodyne G703	wood flour	10% F_2_ + N_2_; 3 h	4400/4800	32.4/41.7	1.4/1.3	[97,133]
EP (10/90)	Kevlar	5% F_2_ + 5% air + He/5% F_2_ + 1% O_2_ + 4% N_2_ + He	400/560/680	20/30/33	6.85/5.13/4.02	[134]
PP	Twaron	5% F_2_ + He; 1.5 h	-	24.74/30.76	4.64/5.84	[24]
PP	Twaron	10% F_2_ + He; 2 h	1144/1480	27.55/31.65	4.6/4.5	[133]
Derakane 411–350 Ashland vinyl ester resin	Kevlar	HF	-	364/115	-	[135]
PP	Illite	F_2_	-	35/38	-	[124]
ED-22	Taunit-M CNTs, 0.1%	F_2_; 150 °C/250 °C	-	77.4/89.6/69.8	-	[113]

^1^ The first value of a parameter corresponds to initial material while other values do to the treated samples.

**Table 4 polymers-12-02836-t004:** Effect of direct fluorination on the surface conductivity $\sigma_{s}$, permittivity ε, DC flashover voltage $V_{f}$ of several polymers and composites in dry air (unless otherwise specified) at room temperature.

Material	$\mathrm{Pristine}\;\sigma_{s}(S\cdot \mathrm{sq})$	$\mathrm{Fluorin}.\;\sigma_{s}(S\cdot \mathrm{sq})$	$\mathrm{Pristine}\;\varepsilon_{st}$	Fluorin. ε	$V_{f},\;\mathrm{kV}$ (*increase in* %)	*Ref.*
PE	-	-	-	-	24.5 (+51%) vacuum	[151]
	-	-	2.4	6	-	[147]
Epoxy resin	5.6 × 10^−18^	2.4 × 10^−14^	-	-	-	[150]
	8 × 10^−18^	4 × 10^−14^	-	-	-	[152]
	6.2 × 10^−20^	1.25 × 10^−15^	-	-	-	[58]
	1.5 × 10^−18^	2.3 × 10^−14^	-	-	91.9 (+12%) SF_4_	[153]
			-	-	27.8 (+26%)	[59]
			-	-	15.1 (+26%)	[154]
			-	-	21.11 (+22%)	[151]
Al_2_O_3_-filled epoxy resin	9.7 × 10^−22^	3.7 × 10^−16^	-	-	-	[57]
	7.7 × 10^−18^	7.9 × 10^−16^	-	-	-	[58]
	8.8 × 10^−19^	3.68 × 10^−16^	-	-	21.1 (+7%) air 28.6 (+4%) SF_4_	[60,61]
	-	-	-	-	25.5 (+16%)	[59]
PDMS	3.7 × 10^−18^	4.4 × 10^−14^	-	-		[54]
			3.6	3.1	15.6(+4%)	[51]
Oil-impregnated paper	5 × 10^−13^	2 × 10^−11^	-	-	-	[155]
Polyimide	-	-	3.3	2.7	-	[48]
	-	-	3.1	2.5	-	[49]
	-	-	3.4	2.8	-	[50]
PTFE	<10^−16^	-	2.1	-	25.3	[156,157]
FEP	<10^−15^	-	2	-	-	
PFA	10^−18^	-	2.03	-	-	

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
