# Peer review of "Direct Fluorination as Method of Improvement of Operational Properties of Polymeric Materials"

_polymers, 2020, doi:10.3390/polym12122836_

Round 1

Reviewer 1 Report

The review article by Belov et al. describes an overview of the post polymer synthesis fluorination of various polymeric materials and their physical, mechanical and electrical properties. Overall the subject matter is unique and gives a reasonably thorough overview of the field. However, there are some issues, mainly in style and organization that must be adjusted before final publication.

The introduction is fine and gives a reasonable overview to introduce the review, and there are only a few minor english inconsistencies. 

Sections 2 and 3 overall are very difficult to follow and need a lot of work to make them flow well. They seem to contain a lot of disconnected paragraphs making it hard to read. Section 4 is much better written, with good review style. It gives credit to the authors referenced by name, gives a good introduction and effectively discusses the data and tables presented. It should be used as a guide for how the other two sections (2 and 3) should be organized.

For section 2, there should be more discussion of the methods of testing the wetability and surface energy. These are referenced, but the discussion of all the properties is lost without first introducing the determination methods. Some intro is given on droplet testing, but for surface energy it is very hard to follow the equations given and where they come from. In Table 1 it is unclear what each of the theta value corresponds (the third entry). In this section a more effective introduction as to what and why needs to be given in the first couple paragraphs before jumping into the data of others. Make sure to also give authors credit by name as done in Intro and Section 4. More pictures could also be helpful with visualization. The jump to free energy from contact angle is also hard to follow and is too intermixed. Maybe break things down by degree of fluorination of the different polymers? And also break down by fluorination method?

In Section 3, I would suggest actually using a form of the closing paragraph as the opening paragraph as opposed to just focusing on applications. It gives a better overview of the properties and importance of the mechanical property differences between fluorinated and non-fluorinated materials. Discussion of Table 3 data? Not really sure of the value of Figure 4.

Section 4 is well written and a good example of a decent review. There are a few minor english errors, but these can be fixed in editorial adjustments. There is a mention of fluorinating silicones. Do the authors of this work see any degradation of the polymers by the fluorination process?

There is one confusing point in the conclusions section. In the second to last line of the conclusions: it says "unfortunately, it is often "utilized"" Do you mean under utilized, as in not used often? Otherwise the "unfortunately" portion is confusing. 

Overall this review can be adjusted to give a reasonable overview of the polymer fluorination field.

Reviewer 2 Report

The article brings an understandable review on polymer materials fluorination. I have only minor formal remarks; mainly, there are a few passages that can be understood in more than single way or are difficult for understanding:

line 65-66: punctuation,

line 82: In addition to used two references, some of classical papers on the usage of wetting method to monitor surface conditions after chemical modifications of surfaces that were published in 20th century might be included (think about refs. 68-69 used just in the following paragraph)

line 96: using of i.e. seems me to be misleading, in my opinion, formulations like ... is the value of an advancing contact angle of a water drop θa2О) on it. or is the value of a contact angle of a water drop on it, more specifically, of advancing contact angle θa2О). might be better
For me personally, indicating material in the subscript makes formula better readable θa(Н2О) if embedded superscripts are possible.

lines 96-97: The sentence can have two meanings; if "to evaluate" is not a purpose, "to (the) evaluation" might be used. According to what it should be said, think whether confine or limit will fit better.

line 107: consider writing p- in italics

line 203: solid polymer/liquid should be in order to be it true.

line 206: and should be instead of и. Knowledge of ... would enable to understand better the sentence if this and not the word change should be the subject.

line 209: decrease: I am not qualified in English, however my dictionary says the decrease of (how much) in (what)

line 219: space missing

line 226: An explanation about double values/ranges in last 3 items might be added

line 380: some punctuation or preposition probably missing, uneasy to understand

lines 460 and 464: If the imposes verb in line 464 belongs to the Applications subject in line 460, both of them should be either in singular, or in plural. Otherwise, the sentence becomes difficult for understanding.

line 476: Are you sure that polymeric based is better than polymer based?

line 493: I appreciate the abbreviations list very much. Ultra high molecular weight polyethylene (UHMWPE) and carbon nanotubes (CNT) mentioned in the article text should be included here; I suggest to check whether there are possibly more materials not included in the list

line 531: Ref. 3 may possibly have DOI 10.2174/1389557515666151016124957 not stated in the list.

line 580: Is not the correct journal of ref. 25 Vysokomolekularnye Soedinenia? (and different page numbers in its version in Polymer Science Ser. B ?) (WOS:A1997XJ11000024)

line 695 ref. 69: the author should be De Gennes, P. G. (P. has been omitted)

line 697 ref. 70: The article title contains polyethylene, not polyethelene. Its should not be capitalised.

line 754 ref. 92: Can omitted DOI be 10.7317/pk.2015.39.2.338 ?
